# Fast Prediction for Large-Scale Kernel Machines

**Cho-Jui Hsieh,  Si Si,  and Inderjit S. Dhillon**
Department of Computer Science
University of Texas at Austin
Austin, TX 78712 USA
`{cjhsieh,ssi,inderjit}@cs.utexas.edu`

## Abstract

Kernel machines such as kernel SVM and kernel ridge regression usually construct high quality models; however, their use in real-world applications remains limited due to the high prediction cost. In this paper, we present two novel insights for improving the prediction efficiency of kernel machines. First, we show that by adding "pseudo landmark points" to the classical Nyström kernel approximation in an elegant way, we can significantly reduce the prediction error without much additional prediction cost. Second, we provide a new theoretical analysis on bounding the error of the solution computed by using Nyström kernel approximation method, and show that the error is related to the weighted kmeans objective function where the weights are given by the model computed from the original kernel. This theoretical insight suggests a new landmark point selection technique for the situation where we have knowledge of the original model. Based on these two insights, we provide a divide-and-conquer framework for improving the prediction speed. First, we divide the whole problem into smaller local subproblems to reduce the problem size. In the second phase, we develop a kernel approximation based fast prediction approach within each subproblem. We apply our algorithm to real world large-scale classification and regression datasets, and show that the proposed algorithm is consistently and significantly better than other competitors. For example, on the Covertype classification problem, in terms of prediction time, our algorithm achieves more than 10000 times speedup over the full kernel SVM, and a two-fold speedup over the state-of-the-art LDKL approach , while obtaining much higher prediction accuracy than LDKL (95.2% vs. 89.53%).

## 1 Introduction

Kernel machines have become widely used in many machine learning problems, including classification, regression, and clustering. By mapping samples to a high-dimensional feature space, kernel machines are able to capture the nonlinear properties and usually achieve better performance compared to linear models. However, computing the decision function for the new test samples is typically expensive which limits the applicability of kernel methods to real-world applications. Therefore speeding up the prediction time of kernel methods has become an important research topic. For example, recently [2, 10] proposed various heuristics to speed up kernel SVM prediction, and kernel approximation based methods [27, 5, 21, 16] can also be applied to speed up the prediction for general kernel machines. Among them, LDKL attracts much attention recently as it performs much better than state-of-the-art kernel approximation and reduced set based methods for fast prediction. Experimental results show that LDKL can reduce the prediction costs by more than three orders of magnitude with little degradation of accuracy as compared with the original kernel SVM.

In this paper, we propose a novel fast prediction technique for large-scale kernel machines. Our method is built on the Nyström approximation, but with the following innovations:

1. We show that by adding "pseudo landmark points" to the Nyström approximation, the kernel approximation error can be reduced without too much additional prediction cost.

2. We provide a theoretical analysis of the model approximation error $\|\bar{\boldsymbol{\alpha}} - \boldsymbol{\alpha}^*\|$, where $\bar{\boldsymbol{\alpha}}$ is the model (solution) computed by Nyström approximation, and $\boldsymbol{\alpha}^*$ is the solution computed from the original kernel. Instead of bounding the error $\|\bar{\boldsymbol{\alpha}} - \boldsymbol{\alpha}^*\|$ by kernel approximation error on the entire kernel matrix, we refine the bound by taking the $\boldsymbol{\alpha}^*$ weights into consideration, which indicates that we only need to focus on approximating the columns in the kernel matrix with large $\boldsymbol{\alpha}^*$ values (e.g., support vectors in kernel SVM problem). We further show that the error bound is connected to the $\boldsymbol{\alpha}^*$-weighted kmeans objective function, which suggests selecting landmark points based on $\boldsymbol{\alpha}^*$ values in Nyström approximation.

3. We consider the above two innovations under a divide-and-conquer framework for fast prediction. The divide-and-conquer framework partitions the problem using kmeans clustering to reduce the problem size, and for each subproblem we apply the above two techniques to develop a kernel approximation scheme for fast prediction.

Based on the above three innovations, we develop a fast prediction scheme for kernel methods, DC-Pred++, and apply it to speed up the prediction for kernel SVM and kernel ridge regression. The experimental results show that our method outperforms state-of-the-art methods in terms of prediction time and accuracy. For example, on the Covertype classification problem, our algorithm achieves a two-fold speedup in terms of prediction time, and yields a higher prediction accuracy (95.2% vs 89.53%) compared to the state-of-the-art fast prediction approach LDKL. Perhaps surprisingly, our training time is usually faster or at least competitive with state-of-the-art solvers.

We begin by presenting related work in Section 2, while the background material is given in Section 3. In Section 4, we introduce the concept of pseudo landmark points in kernel approximation. In Section 5, we present the divide-and-conquer framework, and theoretically analyze using the weighted kmeans to select the landmark points. The experimental results on real-world data are presented in Section 6.

## 2   Related Work

There has been substantial works on speeding up the prediction time of kernel SVMs, and most of the approaches can be applied to other kernel methods such as kernel ridge regression. Most of the previous works can be categorized into the following three types:

**Preprocessing.**   Reducing the size of the training set usually yields fewer support vectors in the model, and thus results in faster prediction speed. [20] proposed a "squashing" approach to reduce the size of training set by clustering and grouping nearby points. [19] proposed to select the extreme points in the training set to train kernel SVM. Nyström method [27, 4, 29] and Random Kitchen Sinks (RKS) [21] form low-rank kernel approximations to improve both training and prediction speed. Although RKS usually requires a larger rank than Nyström method, it can be further sped up by using fast Hadamard transform [16]. Other kernel approximation methods [12, 18, 1] are also proposed for different types of kernels.

**Post-processing.**   Post-processing approaches are designed to reduce the number of support vectors in the testing phase. A comprehensive comparison of these reduced-set methods has been conducted in [11], and results show that the incremental greedy method [22] implemented in STRtool achieves the best performance. Another randomized algorithm to refine the solution of the kernel SVM has been recently proposed in [2].

**Modified Training Process.**   Another line of research aims to reduce the number of support vectors by modifying the training step. [13] proposed a greedy basis selection approach; [24] proposed a Core Vector Machine (CVM) solver to solve the L2-SVM. [9] applied a cutting plane subspace pursuit algorithm to solve the kernel SVM. The Reduced SVM (RSVM) [17] selected a subset of features in the original data, and solved the primal problem of kernel SVM. Locally Linear SVM (LLSVM) [15] represented each sample as a linear combination of its neighbors to yield efficient prediction speed. Instead of considering the original kernel SVM problem, [10] developed a new tree-based local kernel learning model (LDKL), where the decision value of each sample is computed by a series of inner products when traversing the tree.

## 3   Background

**Kernel Machines.**   In this paper, we focus on two kernel machines – kernel SVM and kernel ridge regressions. Given a set of instance-label pairs $\{\boldsymbol{x}_i, y_i\}_{i=1}^n$, $\boldsymbol{x}_i \in \mathbb{R}^d$, the training process of kernel SVM and kernel ridge regression generates $\boldsymbol{\alpha}^* \in \mathbb{R}^n$ by solving the following optimization problems:

$$\text{Kernel SVM: } \boldsymbol{\alpha}^* \leftarrow \operatorname*{argmin}_{\boldsymbol{\alpha}} \frac{1}{2}\boldsymbol{\alpha}^T Q\boldsymbol{\alpha} - \boldsymbol{e}^T\boldsymbol{\alpha} \quad \text{s.t. } 0 \le \boldsymbol{\alpha} \le C, \tag{1}$$

$$\text{Kernel Ridge Regression: } \boldsymbol{\alpha}^* \leftarrow \operatorname*{argmin}_{\boldsymbol{\alpha}} \boldsymbol{\alpha}^T G\boldsymbol{\alpha} + \lambda\boldsymbol{\alpha}^T\boldsymbol{\alpha} - 2\boldsymbol{\alpha}^T\boldsymbol{y}, \tag{2}$$

where $G \in \mathbb{R}^{n \times n}$ is the kernel matrix with $G_{ij} = K(\boldsymbol{x}_i, \boldsymbol{x}_j)$; $Q$ is an $n$ by $n$ matrix with $Q_{ij} = y_i y_j G_{ij}$, and $C, \lambda$ are regularization parameters.

In the prediction phase, the decision value of a testing data $\boldsymbol{x}$ is computed as $\sum_{i=1}^{n} \alpha_i^* K(\boldsymbol{x}_i, \boldsymbol{x})$, which in general requires $O(\bar{n}d)$ where $\bar{n}$ is the number of nonzero elements in $\boldsymbol{\alpha}^*$. Note that for kernel SVM problem, we may think $\alpha_i^*$ is weighted by $y_i$ when computing decision value for $\boldsymbol{x}$. In comparison, linear models only require $O(d)$ prediction time, but usually generate lower prediction accuracy.

**Nyström Approximation.** Kernel machines usually do not scale to large-scale applications due to the $O(n^2 d)$ operations to compute the kernel matrix and $O(n^2)$ space to store it in memory. As shown in [14], low-rank approximation of kernel matrix using the Nyström method provides an efficient way to scale up kernel machines to millions of instances. Given $m \ll n$ landmark points $\{\boldsymbol{u}_j\}_{j=1}^{m}$, the Nyström method first forms two matrices $C \in \mathbb{R}^{n \times m}$ and $W \in \mathbb{R}^{m \times m}$ based on the kernel function, where $C_{ij} = K(\boldsymbol{x}_i, \boldsymbol{u}_j)$ and $W_{ij} = K(\boldsymbol{u}_i, \boldsymbol{u}_j)$, and then approximates the kernel matrix as

$$G \approx \bar{G} := CW^\dagger C^T, \tag{3}$$

where $W^\dagger$ denotes the pseudo-inverse of $W$. By approximating $G$ via Nyström method, the kernel machines are usually transformed to linear machines, which can be solved efficiently. Given the model $\boldsymbol{\alpha}$, in the testing phase, the decision value of $\boldsymbol{x}$ is evaluated as

$$\boldsymbol{c}(W^\dagger C^T \boldsymbol{\alpha}) = \boldsymbol{c}\boldsymbol{\beta},$$

where $\boldsymbol{c} = [K(\boldsymbol{x}, \boldsymbol{u}_1), \dots, K(\boldsymbol{x}, \boldsymbol{u}_m)]$, and $\boldsymbol{\beta} = W^\dagger C^T \boldsymbol{\alpha}$ can be precomputed and stored. To obtain the prediction on one test sample, Nyström approximation only needs $O(md)$ flops to compute $\boldsymbol{c}$, and $O(m)$ flops to compute the decision value $\boldsymbol{c}\boldsymbol{\beta}$, so it becomes an effective ways to improve the prediction speed. However, Nyström approximation usually needs more than 100 landmark points to achieve reasonable good accuracy, which is still expensive for large-scale applications.

## 4 Pseudo Landmark Points for Speeding up Prediction Time

In Nyström approximation, there is a trade-off in selecting the number of landmark points $m$. A smaller $m$ means faster prediction speed, but also yields higher kernel approximation error, which results in a lower prediction accuracy. Therefore we want to tackle the following problem – *can we add landmark points without increasing the prediction time?*

Our solution is to construct extra "pseudo landmark points" for the kernel approximation. Recall that originally we have $m$ landmark points $\{\boldsymbol{u}_j\}_{j=1}^{m}$, and now we add $p$ pseudo landmark points $\{\boldsymbol{v}_t\}_{t=1}^{p}$ to this set. In this paper, we consider pseudo landmark points are sampled from the training dataset, while in general each pseudo landmark point can be any $d$-dimensional vector. The only difference between pseudo landmark points and landmark points is that the kernel values $K(\boldsymbol{x}, \boldsymbol{v}_t)$ are computed in a fast but approximate manner in order to speed up the prediction time. We use a regression-based method to approximate $\{K(\boldsymbol{x}, \boldsymbol{v}_t)\}_{t=1}^{p}$. Assume for each pseudo landmark point $\boldsymbol{v}_t$, there exists a function $f_t : \mathbb{R}^m \to \mathbb{R}$, where the input to each $f_t$ is the computed kernel values $\{K(\boldsymbol{x}, \boldsymbol{u}_j)\}_{j=1}^{m}$, and the output is an estimator of $K(\boldsymbol{x}, \boldsymbol{v}_t)$. We can either design the function for specific kernels, for example, in Section 4.1 we design $f_t$ for stationary kernels, or learn $f_t$ by regression for general kernels (Section 4.2).

Before introducing the design or learning process for $\{f_t\}_{t=1}^{p}$, we first describe how to use them to form the Nyström approximation. With $p$ pseudo landmark points and $\{f_t\}_{t=1}^{p}$ given, we can form the following a $n \times (m + p)$ matrix $\bar{C}$, by adding the $p$ extra columns to $C$:

$$\bar{C} = [C, \ C'], \ \text{where } C'_{it} = f_t(\{K(\boldsymbol{x}_i, \boldsymbol{u}_j)\}_{j=1}^{m}) \ \forall i = 1, \dots, n \text{ and } \forall t = 1, \dots, p. \tag{4}$$

Then the kernel matrix $G$ can be approximated by

$$G \approx \bar{G} = \bar{C}\bar{W}\bar{C}^T, \ \text{with } \bar{W} = \bar{C}^\dagger G(\bar{C}^\dagger)^T, \tag{5}$$

where $\bar{C}^\dagger$ is the pseudo inverse of $\bar{C}$; $\bar{W}$ is the optimal solution to minimize $\|G - \bar{G}\|_F$ if $\bar{G}$ is restricted to the range space of $\bar{C}$, which is also used in [26]. Note that in our case $\bar{W}$ cannot be

obtained by inverting an $m + p$ by $m + p$ matrix as in the original Nyström approach in (3), because the kernel values between $x$ and pseudo landmark points are the approximate kernel values. As a result the time to form the Nyström approximation in (5) is slower than forming (3) since the whole kernel matrix $G$ has to be computed.

If the number of samples $n$ is too large to compute $G$, we can estimate the matrix $\bar{W}$ by minimizing the approximation error on a submatrix of $G$. More specifically, we randomly select a submatrix $G_{\mathrm{sub}}$ from $G$ with row/and column indexes $\mathcal{I}$. If we focus on approximating $G_{\mathrm{sub}}$, the optimal $\bar{W}$ is $\bar{W} = (\bar{C}_{\mathcal{I},:})^{\dagger} G_{\mathrm{sub}} ((\bar{C}_{\mathcal{I},:})^{\dagger})^T$, which only requires computation of $O(|\mathcal{I}|^2)$ kernel elements.

Based on the approximate kernel $\bar{G}$, we can train a model $\bar{\alpha}$ and store the vector $\bar{\beta} = \bar{W} \bar{C}^T \bar{\alpha}$ in memory. For a testing sample $x$, we first compute the kernel values between $x$ and landmarks points $\boldsymbol{c} = [K(\boldsymbol{x}, \boldsymbol{u}_1), \dots, K(\boldsymbol{x}, \boldsymbol{u}_m)]$, which usually requires $O(md)$ flops, and then expand $\boldsymbol{c}$ to an $(m + p)$-dimensional vector $\bar{\boldsymbol{c}} = [\boldsymbol{c}, f_1(\boldsymbol{c}), \dots, f_p(\boldsymbol{c})]$ based on the $p$ pseudo landmark points and the functions $\{f_t\}_{t=1}^p$. Assume each $f_t(\boldsymbol{c})$ function can be evaluated with $O(s)$ time, then we can easily compute $\bar{\boldsymbol{c}}$ and the decision value $\bar{\boldsymbol{c}}\bar{\beta}$ taking $O(md + ps)$ time, where $s$ is much smaller than $d$. Overall, our algorithm can be summarized in Algorithm 1.

---

**Algorithm 1:** Kernel Approximation with Pseudo Landmark Points

---

**Kernel Approximation Steps:**
    Select $m$ landmark points $\{\boldsymbol{u}_j\}_{j=1}^m$.
    Compute $n \times m$ matrix $C$ where $C_{ij} = K(\boldsymbol{x}_i, \boldsymbol{u}_j)$.
    Select $p$ pseudo landmark points $\{\boldsymbol{v}_t\}_{t=1}^p$.
    Construct $p$ functions $\{f_t\}_{t=1}^p$ by methods in Section 4.1 or Section 4.2.
    Expand $C$ to $\bar{C}$ as $\bar{C} = [C,\ C']$ by (4), and compute $\bar{W}$ by (5).
**Training:** Compute $\bar{\alpha}$ based on $\bar{G}$ and precompute $\bar{\beta} = \bar{W} \bar{C}^T \bar{\alpha}$.
**Prediction for a test point $x$:**
    Compute $m$ dimensional vector $\boldsymbol{c} = [K(\boldsymbol{x}, \boldsymbol{u}_1), \dots, K(\boldsymbol{x}, \boldsymbol{u}_m)]$.
    Compute $m + p$ dimensional vector $\bar{\boldsymbol{c}} = [\boldsymbol{c}, f_1(\boldsymbol{c}), \dots, f_p(\boldsymbol{c})]$.
    Decision value: $\bar{\boldsymbol{c}}\bar{\beta}$.

---

## 4.1 Design the functions for stationary kernels

Next we discuss various ways to design/learn the functions $\{f_t\}_{t=1}^p$. First we consider the stationary kernels $K(\boldsymbol{x}, \boldsymbol{v}_t) = \kappa(\|\boldsymbol{x} - \boldsymbol{v}_t\|)$, where the kernel approximation problem can be reduced to estimate $\|\boldsymbol{x} - \boldsymbol{v}_t\|$ with low cost. Suppose we choose $p$ pseudo landmark points $\{\boldsymbol{v}_t\}_{t=1}^p$ by randomly sampling $p$ points in the dataset. By the triangle inequality,

$$\max_j \left( |\,\|\boldsymbol{x} - \boldsymbol{u}_j\| - \|\boldsymbol{v}_t - \boldsymbol{u}_j\|\,| \right) \leq \|\boldsymbol{x} - \boldsymbol{v}_t\| \leq \min_j \left( \|\boldsymbol{x} - \boldsymbol{u}_j\| + \|\boldsymbol{v}_t - \boldsymbol{u}_j\| \right). \qquad (6)$$

Since $\|\boldsymbol{x} - \boldsymbol{u}_j\|$ has already been evaluated for all $\boldsymbol{u}_j$ (to compute $K(\boldsymbol{x}, \boldsymbol{u}_j)$) and $\|\boldsymbol{v}_t - \boldsymbol{u}_j\|$ can be precomputed, we can use either left hand side or right hand side of (6) to estimate $K(\boldsymbol{x}, \boldsymbol{v}_t)$. We can see that approximating $K(\boldsymbol{x}, \boldsymbol{v}_t)$ using (6) only requires $O(m)$ flops and is more efficient than computing $K(\boldsymbol{x}, \boldsymbol{v}_t)$ from scratch when $m \ll d$ ($d$ is the dimensionality of data).

## 4.2 Learning the functions for general kernels

Next we consider learning the function $f_t$ for general kernels by solving a regression problem. Assume each $f_t$ is a degree-$D$ polynomial function (in the paper we only use $D = 2$). Let $\mathcal{Z}$ denote the basis functions: $\mathcal{Z} = \{(i_1, \dots, i_m) \mid i_1 + \dots + i_m = d\}$, and for each element $z^{(q)} \in \mathcal{Z}$ we denote the corresponding polynomial function as $Z^{(q)}(\boldsymbol{c}) = c_1^{z_1^{(q)}} c_2^{z_2^{(q)}} \dots c_m^{z_m^{(q)}}$. Each $f_t$ can then be written as $f_t(\boldsymbol{c}) = \sum_q a_q^t Z^{(q)}(\boldsymbol{c})$. A naive way to apply the pseudo-landmark technique using polynomial functions is: to learn the optimal coefficients $\{a_q^t\}_{q=1}^{|\mathcal{Z}|}$ for each $t$, and then compute $\bar{C}, \bar{W}$ based on (4) and (5). However, this two-step procedure requires a huge amount of training time, and the prediction time cannot be improved if $|\mathcal{Z}|$ is large.

Therefore, we consider implicitly applying the pseudo-landmark point technique. We expand $C$ by

$$\hat{C} = [C,\ C''], \text{ where } C_{iq}'' = Z^{(q)}(\boldsymbol{c}_i). \qquad (7)$$

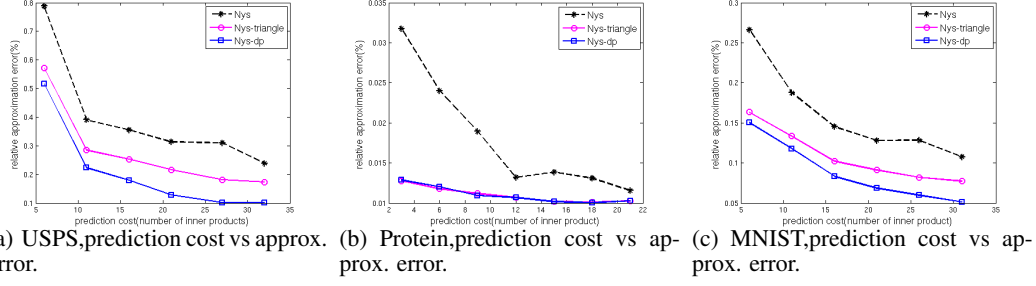

| (a) USPS,prediction cost vs approx. error. | (b) Protein,prediction cost vs approx. error. | (c) MNIST,prediction cost vs approx. error. |

Figure 1: Comparison of different pseudo landmark points strategy. The relative approximation error is $\|G-\bar{G}\|_F/\|G\|_F$ where $G$ and $\bar{G}$ is the real and approximate kernel respectively. We observe that both Nys-triangle (using the triangular inequality to approximate kernel values) and Nys-dp (using the polynomial expansion with the degree $D=2$) can dramatically reduce the approximation error under the same prediction cost.

where $\boldsymbol{c}_i = [K(\boldsymbol{x}_i, \boldsymbol{u}_1), \dots, K(\boldsymbol{x}_i, \boldsymbol{u}_m)]$ and each $Z^{(q)}(\cdot)$ is the $q$-th degree-$D$ polynomial basis with $q = 1, \dots, |\mathcal{Z}|$. After forming $\hat{C}$, we can then compute $\hat{W} = \hat{C}^\dagger G(\hat{C}^\dagger)^T$ and approximate the kernel by $\hat{C}\hat{W}\hat{C}^T$. This procedure is much more efficient than the previous two-step procedure where we need to learn $\{a_q^t\}_{q=1}^{|\mathcal{Z}|}$, and more importantly, in the following lemma we show that this approach gives better approximation to the previous two-step procedure.

**Lemma 1.** *If $\{f_t(\cdot)\}_{t=1}^p$ are degree-$D$ polynomial functions, $\bar{C}, \bar{W}$ are computed by (4), (5) and $\hat{C}, \hat{W}$ are computed by (7), (5), then $\|G - \bar{C}\bar{W}\bar{C}^T\| \geq \|G - \hat{C}\hat{W}\hat{C}^T\|$.*

The proof is in Appendix 7.3. In practice we do not need to form all the low degree polynomial basis – just sample some of the basis from $\mathcal{Z}$ is enough. Figure 1 compares using Nyström method with or without pseudo landmark points for approximating Gaussian kernels. For each dataset, we choose a few number of landmark points (2-30), and add pseudo landmark points according the triangular inequality (6) or according to the polynomial function (7). We observe that the kernel approximation error is dramatically reduced under the same prediction cost. Note that we can also apply this pseudo-landmark points approach as a building block in other kernel approximation frameworks, e.g., the Memory Efficient Kernel Approximation (MEKA) proposed in [23].

## 5 Weighted Kmeans Sampling with a Divide-and-Conquer Framework

In all the related work, Nyström approximation is considered as a preprocessing step, which does not incorporate the information from the model itself. In this section, we consider the case that the model $\boldsymbol{\alpha}^*$ for kernel SVM or kernel ridge regression is given, and derive a better approach to select landmark points. The approach can be used in conjunction with divide-and-conquer SVM [8] where an approximate solution to $\boldsymbol{\alpha}^*$ can be computed efficiently.

Let $\boldsymbol{\alpha}^*$ be the optimal solution of the kernel machines computed with the original kernel matrix $G$, and $\bar{\boldsymbol{\alpha}}$ be the approximate solution by using approximate kernel matrix $\bar{G}$. We derive the following upper bound of $\|\bar{\boldsymbol{\alpha}} - \boldsymbol{\alpha}^*\|$ for both kernel SVM and kernel ridge regression:

**Theorem 1.** *Let $\boldsymbol{\alpha}^*$ be the optimal solution for kernel ridge regression with kernel matrix $G$, and $\bar{\boldsymbol{\alpha}}$ is the solution for kernel ridge regression with kernel $\bar{G}$ obtained by Nyström approximation (3), then*

$$\|\bar{\boldsymbol{\alpha}} - \boldsymbol{\alpha}^*\| \leq \Delta/\lambda \quad \text{with } \Delta = \sum_{i=1}^n |\alpha_i^*| \|\bar{G}_{\cdot,i} - G_{\cdot,i}\|,$$

*where $\lambda$ is the regularization parameter in kernel ridge regression, and $\bar{G}_{\cdot,i}$ and $G_{\cdot,i}$ are the $i$-th column of $\bar{G}$ and $G$ respectively.*

**Theorem 2.** *Let $\boldsymbol{\alpha}^*$ be the optimal solution for kernel SVM with kernel $G$, and $\bar{\boldsymbol{\alpha}}$ be the solution of kernel SVM with kernel $\bar{G}$ obtained by Nyström approximation (3), then*

$$\|\bar{\boldsymbol{\alpha}} - \boldsymbol{\alpha}^*\| \leq \theta^2 \|W\|_2 (1 + \rho)\Delta, \tag{8}$$

*where $\rho$ is the largest eigenvalue of $\bar{G}$, and $\theta$ is a positive constant independent on $\boldsymbol{\alpha}^*, \bar{\boldsymbol{\alpha}}$.*

The proof is in Appendix 7.4 and 7.5. Here we show that $\|\bar{\boldsymbol{\alpha}} - \bar{\boldsymbol{\alpha}}^*\|$ can be upper bounded by a weighted kernel approximation error. This result looks natural but has a significant consequence – to get a good approximate model, we do not need to minimize the kernel approximation error on all the $n^2$ elements of $G$; instead, the quality of solution is mostly affected by a small portion of columns of $G$ with larger $|\alpha_i^*|$. For example, in the kernel SVM problem, $\boldsymbol{\alpha}^*$ is a sparse vector containing many zero elements, and the above bound indicates that we just need to approximate the columns in $G$ with corresponding $\alpha_i^* \neq 0$ accurately. Based on the error bounds, we want to select landmark points for Nyström approximation that minimize $\Delta$. We focus on the kernel functions that satisfy

$$(K(\boldsymbol{a}, \boldsymbol{b}) - K(\boldsymbol{c}, \boldsymbol{d}))^2 \leq C_K (\|\boldsymbol{a} - \boldsymbol{c}\|^2 + \|\boldsymbol{b} - \boldsymbol{d}\|^2), \forall \boldsymbol{a}, \boldsymbol{b}, \boldsymbol{c}, \boldsymbol{d}, \tag{9}$$

where $C_K$ is a kernel-dependent constant. It has been shown in [29] that all the stationary kernels $(K(\boldsymbol{x}_i, \boldsymbol{x}_j) = \kappa(\|\boldsymbol{x}_i - \boldsymbol{x}_j\|))$ satisfy (9). Next we show that the weighted kernel approximation error $\Delta$ is upper bounded by the weighted kmeans objective.

**Theorem 3.** *If the kernel function satisfies condition* (9), *and let* $\boldsymbol{u}_1, \ldots, \boldsymbol{u}_m$ *be the landmark points for constructing the Nyström approximation* ($\bar{G} = CW^\dagger C^T$), *then*

$$\Delta \leq (n + n\|W^\dagger\| \sqrt{k \gamma_{max}}) \sqrt{C_k} \sqrt{D_{\boldsymbol{\alpha}^*}^2 (\{\boldsymbol{u}_j\}_{j=1}^m)},$$

*where* $\gamma_{max}$ *is the upper bound of kernel function,*

$$D_{\boldsymbol{\alpha}}^2 (\{\boldsymbol{u}_i\}_{i=1}^m) := \sum_{i=1}^n \alpha_i^2 \|\boldsymbol{x}_i - \boldsymbol{u}_{\pi(i)}\|^2, \tag{10}$$

*and* $\pi(i) = \arg\min_s \|\boldsymbol{u}_s - \boldsymbol{x}_i\|^2$ *is the landmark point closest to* $\boldsymbol{x}_i$.

The proof is in Appendix 7.6. Note that $D_{\boldsymbol{\alpha}^*}^2 (\{\boldsymbol{u}_i\}_{i=1}^m)$ is the weighted kmeans objective function with $\{(\alpha_i^*)^2\}_{i=1}^n$ as the weights. Combining Theorems 1, 2, and 3, we conclude that for both kernel SVM and ridge regression, the approximation error $\|\bar{\boldsymbol{\alpha}} - \boldsymbol{\alpha}^*\|$ can be upper bounded by the weighted kmeans objective function. As a consequence, if $\boldsymbol{\alpha}^*$ is given, we can use weighted kmeans with weights $\{(\alpha_i^*)^2\}_{i=1}^n$ to find the landmark points $\boldsymbol{u}_1, \ldots, \boldsymbol{u}_m$, which tends to minimize the approximation error. In Figure 4 (in the Appendix) we show that for the kernel SVM problem, selecting landmark points by weighted kmeans is a very effective strategy for fast and accurate prediction in real-world datasets.

In practice we do not know $\boldsymbol{\alpha}^*$ before training the kernel machines, and exactly computing $\boldsymbol{\alpha}^*$ is very expensive for large-scale datasets. However, using weighted kmeans to select landmark points can be combined with any approximate solvers – we can use an approximate solver to quickly approximate $\boldsymbol{\alpha}^*$, and then use it as the weights for the weighted kmeans. Next we show how to combine this approach with the divide-and-conquer framework recently proposed in [8, 7].

**Divide and Conquer Approach.** The divide-and-conquer SVM (DC-SVM) was proposed in [8] to solve the kernel SVM problem. The main idea is to divide the whole problem into several smaller subproblems, where each subproblem can be solved independently and efficiently. [8] proposed to partition the data points by kernel clustering, but this approach is expensive in terms of prediction efficiency. Therefore we use kmeans clustering in the input space to build the hierarchical clustering.

Assume we have $k$ clusters as the leaf nodes, the DC-SVM algorithm computes the solutions $\{(\boldsymbol{\alpha}^{(i)})^*\}_{i=1}^k$ for each cluster independently. For a testing sample, they use an "early prediction" scheme, where the testing sample is first assigned to the nearest cluster and then the local model in that cluster is used for prediction. This approach can reduce the prediction time because it only computes the kernel values between the testing sample and all the support vectors in one cluster. However, the model in each cluster may still contain many support vectors, so we propose to approximate the kernel in each cluster by Nyström based kernel approximation as mentioned in Section 4 to further reduce the prediction time. In the prediction step we first go through the hierarchical tree to identify the nearest cluster, and then compute the kernel values between the testing sample and the landmark points in that cluster. Finally, we can compute the decision value based on the kernel values and the prediction model. The same idea can be applied to kernel ridge regression. Our overall algorithm – DC-Pred++ is presented in Algorithm 2.

## 6  Experimental Results

In this section, we compare our proposed algorithm with other fast prediction algorithms for kernel SVM and kernel ridge regression problems. All the experiments are conducted on a machine with

**Algorithm 2:** DC-Pred++: our proposed divide-and-conquer approach for fast Prediction.
___
**Input**  : Training samples $\{\boldsymbol{x}_i\}_{i=1}^n$, kernel function $K$.
**Output**: A fast prediction model.
**Training:**
   Construct a hierarchical clustering tree with $k$ leaf nodes by kmeans.
   Compute local models $\{(\boldsymbol{\alpha}^{(i)})^*\}_{i=1}^k$ for each cluster.
   For each cluster, use weighted kmeans centroids as landmark points.
   For each cluster, run the proposed kernel approximation with pseudo landmark points
   (Algorithm 1) and use the approximate kernel to train a local prediction model.
**Prediction on $\boldsymbol{x}$:**
   Identify the nearest cluster.
   Run the prediction phase of Algorithm 1 using local prediction models.
___

Table 1: Comparison of kernel SVM prediction on real datasets. Note that the actual prediction time is normalized by the linear prediction time. For example, 12.8x means the actual prediction time = $12.8\times$ (time for linear SVM prediction time).

| Dataset | Metric | DC-Pred++ | LDKL | kmeans Nyström | AESVM | STPRtool | Fastfood |
|---|---|---|---|---|---|---|---|
| Letter | Prediction Time | **12.8x** | 29x | 140x | 1542x | 50x | 50x |
| $n_{\text{train}} = 12,000,$ | Accuracy | **95.90%** | 95.78% | 87.58% | 80.97% | 85.9% | 89.9% |
| $n_{\text{test}} = 6,000, d = 16$ | Training Time | **1.2s** | 243s | 3.8s | 55.2s | 47.7s | 15s |
| CovType | Prediction Time | **18.8x** | 35x | 200x | 3157x | 50x | 60x |
| $n_{\text{train}} = 522,910,$ | Accuracy | **95.19%** | 89.53% | 73.63% | 75.81% | 82.14% | 66.8% |
| $n_{\text{test}} = 58,102, d = 54$ | Training Time | 372s | 4095s | 1442s | **204s** | 77400s | 256s |
| Usps | Prediction Time | 14.4x | **12.01x** | 200x | 5787x | 50x | 80x |
| $n_{\text{train}} = 7291,$ | Accuracy | 95.56% | **95.96%** | 92.53% | 85.97% | 93.6% | 94.39% |
| $n_{\text{test}} = 2007, d = 256$ | Training Time | **2s** | 19s | 4.8s | 55.3s | 34.5s | 12s |
| Webspam | Prediction Time | **20.5x** | 23x | 200x | 4375x | 50x | 80x |
| $n_{\text{train}} = 280,000,$ | Accuracy | **98.4%** | 95.15% | 95.01% | **98.4%** | 91.6% | 96.7% |
| $n_{\text{test}} = 70,000, d = 254$ | Training Time | 239s | 2158s | **181s** | 909s | 32571s | 1621s |
| Kddcup | Prediction Time | **11.8x** | 26x | 200x | 604x | 50x | 80x |
| $n_{\text{train}} = 4,898,431,$ | Accuracy | **92.3%** | 92.2% | 87% | 92.1% | 89.8% | 91.1% |
| $n_{\text{test}} = 311,029, d = 134$ | Training Time | 154s | 997s | 1481s | 2717s | 4925s | 970s |
| a9a | Prediction Time | **12.5x** | 32x | 50x | 4859x | 50x | 80 |
| $n_{\text{train}} = 32,561,$ | Accuracy | **83.9%** | 81.95% | 83.9% | 81.9% | 82.32% | 61.9% |
| $n_{\text{test}} = 16,281, d = 123$ | Training Time | 6.3s | 490s | **1.28s** | 33.17s | 69.1s | 59.9s |

an Intel 2.83GHz CPU with 32G RAM. Note that the prediction cost is shown as actual prediction time dividing by the linear model's prediction time. This measurement is more robust to the actual hardware configuration and provides a comparison with the linear methods.

## 6.1 Kernel SVM

We use six public datasets (shown in Table 1) for the comparison of kernel SVM prediction time. The parameters $\gamma, C$ are selected by cross validation, and the detailed description of parameters for other competitors are shown in Appendix 7.1. We compare with the following methods:

1. DC-Pred++: Our proposed framework, which involves Divide-and-Conquer strategy and applies weighted kmeans to select landmark points and then uses these landmark points to generate pseudo-landmark points in Nyström approximation for fast prediction.
2. LDKL: The Local Deep Kernel Learning method proposed in [10]. They learn a tree-based primal feature embedding to achieve faster prediction speed.
3. Kmeans Nyström: The Nyström approximation using kmeans centroids as landmark points [29]. The resulting linear SVM problem is solved by LIBLINEAR [6].
4. AESVM: Approximate Extreme points SVM solver proposed in [19]. It uses a preprocessing step to filter out unimportant points to get a smaller model.
5. Fastfood: Random Hadamard features for kernel approximation [16].
6. STPRtool: The kernel computation toolbox that implemented the reduced-set post processing approach using the greedy iterative solver proposed in [22].

Note that [10] reported that LDKL achieves much faster prediction speed compared with Locally Linear SVM [15], and reduced set methods [9, 3, 13], so we omit their comparisons here.

The results presented in Table 1 show that DC-Pred++ achieves the best prediction efficiency and accuracy in 5 of the 6 datasets. In general, DC-Pred++ takes less than half of the prediction time and

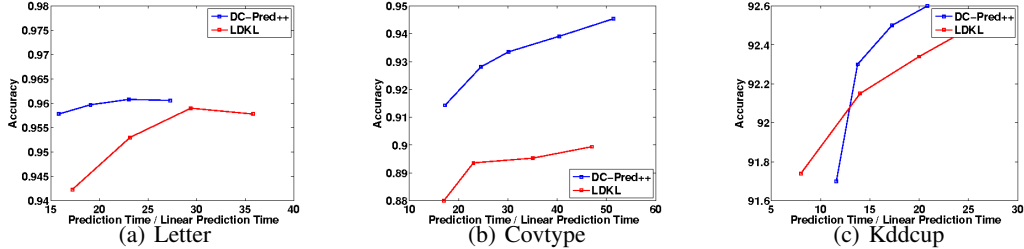

| (a) Letter | (b) Covtype | (c) Kddcup |

Figure 2: Comparison between our proposed method and LDKL for fast prediction in kernel SVM problem. x-axis is the prediction cost and y-axis shows the prediction accuracy. For results on more datasets, please see Figure 5 in the Appendix.

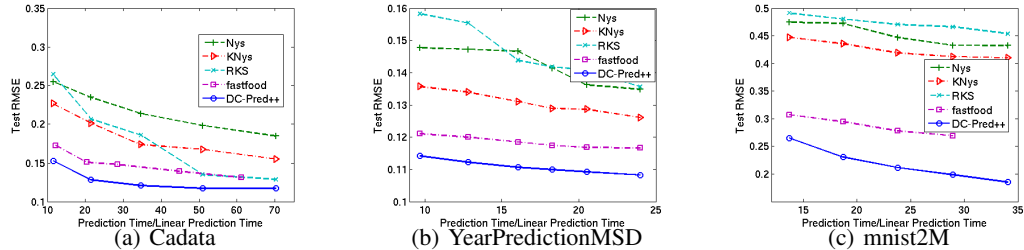

| (a) Cadata | (b) YearPredictionMSD | (c) mnist2M |

Figure 3: Kernel ridge regression results for various datasets. x-axis is the prediction cost and y-axis shows the Test RMSE. All the results are averaged over five independent runs. For results on more datasets, please see Figure 7 in the Appendix.

can still achieve better accuracy than LDKL. Interestingly, in terms of the training time, DC-Pred++ is almost 10 times faster than LDKL on most of the datasets. Since LDKL is the most competitive method, we further show the comparison with LDKL by varying the prediction cost in Figure 2. The results show that on 5 datasets DC-Pred++ achieves better prediction accuracy using the same prediction time.

Note that our approach is an improvement over the divide-and-conquer SVM (DC-SVM) proposed in [8], therefore we further compare DC-Pred++ with DC-SVM in Appendix 7.8. The results clearly demonstrate that DC-Pred++ achieves faster prediction speed, and the main reason is due to the two innovations presented in this paper – adding pseudo landmark points and weighted kmeans to select landmark points to improve Nyström approximation. Finally, we also present the trade-off of two parameters in our algorithm, number of clusters and number of landmark points, in Appendix 7.9.

Table 2: Dataset statistics

| dataset | Cpusmall | Cadata | Census | YearPredictionMSD | mnist2M |
|---|---|---|---|---|---|
| $n_{\text{train}}$ | 6,553 | 16,521 | 18,277 | 463,715 | 1,500,000 |
| $n_{\text{test}}$ | 1,639 | 4,128 | 4,557 | 51,630 | 500,000 |
| $d$ | 12 | 137 | 8 | 90 | 800 |

## 6.2 Kernel Ridge Regression

We further demonstrate the benefits of DC-Pred++ for fast prediction in kernel ridge regression problem on five public datasets listed in Table 2. Note that for mnist2M, we perform regression on two digits and set the target variables to be 0 and 1. We compare DC-Pred++ with other four state-of-the-art kernel approximation methods for kernel ridge regression including the standard Nystrom(Nys)[5], Kmeans Nystrom(KNys)[28], Random Kitchen Sinks(RKS)[21], and Fastfood [16]. All experimental results are based on Gaussian kernel. It is unclear how to generalize LDKL for kernel ridge regression, so we do not compare with LDKL here. The parameters used are chosen by five fold cross-validation (see Appendix 7.1). Figure 3 presents the Test RMSE(root mean squared error on the test data) by varying the prediction cost. To control the prediction cost, for Nys, KNys, and DC-Pred++, we vary the number of landmark points, and for RKS and fastfood, we vary the number of random features. In Figure 3, we can observe that with the same prediction cost, DC-Pred++ always yields lower Test RMSE than other methods.

### Acknowledgements

This research was supported by NSF grants CCF-1320746 and CCF-1117055. C.-J.H also acknowledges support from an IBM PhD fellowship.

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
