[Supplementary Material]

# 7 Appendix

## 7.1 Parameters and Settings

For DC-Pred++, LDKL, and Kmeans Nyström, we choose the parameters $\gamma$ and $C$ for Gaussian kernel by cross validation on training data. For LDKL, we use exactly the same parameters in their public available code for Letter, CovType, and Usps datasets, and conduct cross validation on $W, \lambda_\theta, \lambda_{\theta'}, \sigma$ for other three datasets according to the range given in their website. To control the prediction time of LDKL, we select different numbers of layers in the algorithm. For AESVM, we follow the experimental setting in [19] and use $P = 10000, V = 100$. To control the prediction time for AESVM, we set different thresholds $\epsilon$ in their algorithm. For DC-Pred++, we apply a hierarchical clustering for 10 levels (1024 clusters on the bottom level), and use Divide-and-Conquer SVM [8] to train the SVM classifier from bottom to up until the number of support vectors is larger than 400, and then apply MEKA with 4 clusters with the proposed Nyström expansion technique on each subproblem (so each cluster has approximately $m = 100$ support vectors). We then choose $rm$ landmark points (by weighted kmeans) with different choices of $r < 1$ (ratio) to control the prediction time. We use kernel expansion to get $2rm$ "pseudo landmark points".

For the datasets: Letter, CovType, and Usps are directly downloaded from the LDKL website[10]. The Webspam and a9a datasets are downloaded from LIBSVM tools, and Kddcup is from the UCI data repository. We use a random 80%-20% split for Webspam and Kddcup, and the default splitting for a9a. For more details, please see the following setting.

1. For all the kmeans algorithm, we run 15 steps.
2. For AESVM, we set $P = 10000, V = 1000$ as suggested by their paper.
3. For Kmeans Nyström, we use dual coordinate descent solver in Liblinear.
4. Letter dataset: $\gamma = 2^{-4}$, $C = 8$, AESVM $\epsilon = 1$, LDKL $D = 9, \lambda_W = 0.1, \lambda_\theta = 0.01, \lambda_{\theta'} = 0.001, \sigma = 1$; DC-Pred++ $k = 64$.
5. Covtype dataset: $\gamma = 16, C = 32$, DC-Pred++ $k = 1024$, LDKL: $D = 11, \lambda_W = 0.01, \lambda_\theta = 0.01, \lambda_{\theta'} = 0.001, \sigma = 1$; AESVM $\epsilon = 4$.
6. Usps dataset: $\gamma = 2^{-7}, C = 2^3$, DC-Pred++ $k = 32$, AESVM $\epsilon = 1$, LDKL $D = 3, \lambda_W = 0.1, \lambda_\theta = 0.01, \lambda_{\theta'} = 0.01, \sigma = 0.1$.
7. Webspam dataset: $\gamma = 2^3, C = 2^3$, DC-Pred++ $k = 128$, AESVM $\epsilon = 0.1$, LDKL $D = 9, \lambda_W = 0.0001, \lambda_\theta = 0.001, \lambda_{\theta'} = 0.1, \sigma = 0.1$.
8. Kddcup dataset: $\gamma = 2^{-1}, C = 2^8$, DC-Pred++ $k = 32$,
9. a9a dataset: $\gamma = 2^{-3}, C = 1$, DC-Pred++ $k = 128$, AESVM $\epsilon = 0.7$, LDKL $D = 7, \lambda_W = 0.1, \lambda_\theta = 0.1, \lambda_{\theta'} = 0.01, \sigma = 0.01$. AESVM $\epsilon = 0.5$.
10. Cpusmall dataset: $\gamma = 2^{-4}$, $\lambda = 4$, $k = 5$.
11. Cadata dataset: $\gamma = 2^{-4}$, $\lambda = 2^{-5}$, $k = 5$.
12. Census dataset: $\gamma = 2^2$, $\lambda = 2^{-3}$, $k = 5$.
13. YearPredictionMSD dataset: $\gamma = 0.5$, $\lambda = 2^{-4}$, $k = 40$.
14. minst2M dataset: $\gamma = 2^{-5}$, $\lambda = 5^{-2}$, $k = 40$.

## 7.2 Comparisons of Using Weighted Kmeans and Kmeans to Select Landmark Points

As discussed in Section 5, given the optimal solution $\boldsymbol{\alpha}^*$, the weighted kmeans can be used to select landmark points of the Nyström approximation. Here we show an experimental comparison between using the weighted kmeans and kmeans (which does not incorporate the information from $\boldsymbol{\alpha}^*$) to select landmark points. Figure 4 clearly shows that using weighted kmeans centroids as landmark points achieves better prediction accuracy under the same prediction time on the kernel SVM problem.

## 7.3 Proof of Lemma 1

Given the columns $\hat{C}$, the matrix $\hat{W} = \hat{C}^\dagger G (\hat{C}^\dagger)^T$ is the optimal solution of

$$\underset{W}{\operatorname{argmin}} \|G - \hat{C} W \hat{C}^T\|_F^2.$$

(a) USPS, prediction cost vs prediction accuracy.

Figure 4: A comparison between weighted kmeans landmark points and kmeans landmark points for Nyström approximation. Using weighted kmeans, we can achieve a better prediction error using less number of landmark points. This comparison is based on DC-Pred++ with/without using $\boldsymbol{\alpha}^*$-weighted kmeans.

Therefore, it suffices to show span$(\bar{C}) \subseteq$ span$(\hat{C})$ in order to prove this lemma. By definition $\bar{C} = [C, \ C']$ and $C'_{it} = f_t(\boldsymbol{c}_i)$, where $\boldsymbol{c}_i$ is the $i$-th row of $C$. Since $f_t$ is a polynomial function, it can be written as

$$C'_{it} = \sum_{q=1}^{|\mathcal{Z}|} a_t^q Z^{(q)}(\boldsymbol{c}), \ \ \forall i, t,$$

and this can be further rewritten as $C' = C''A$ where $A$ contains the coefficients $a_t^q$ in the above equation. This implies span$(\bar{C}) \subseteq$ span$(\hat{C})$, which proves the lemma.

### 7.4 Proof of Theorem 1

*Proof.* We define $f_{rg}(\boldsymbol{\alpha})$ to be the dual objective function of kernel ridge regression with kernel $G$ (as defined in (2)) and $\bar{f}_{rg}(\boldsymbol{\alpha})$ to be the objective function with approximate kernel $\bar{G}$. Let $\boldsymbol{\alpha}^*$ and $\bar{\boldsymbol{\alpha}}$ be the optimal solutions for $f_{rg}(\boldsymbol{\alpha})$ and $\bar{f}_{rg}(\boldsymbol{\alpha})$ respectively. Taking the gradient of $f_{rg}(\boldsymbol{\alpha})$, we have $\nabla f_{rg}(\boldsymbol{\alpha}) = 2G\boldsymbol{\alpha} + 2\lambda\boldsymbol{\alpha} - 2\boldsymbol{y}$. Thus

$$\begin{aligned}
\|\nabla \bar{f}_{rg}(\boldsymbol{\alpha}^*) - \nabla \bar{f}_{rg}(\bar{\boldsymbol{\alpha}})\| &= \|2\bar{G}\boldsymbol{\alpha}^* + 2\lambda\boldsymbol{\alpha}^* - 2\boldsymbol{y}\| \\
&= \|2\bar{G}\boldsymbol{\alpha}^* - 2G\boldsymbol{\alpha}^* + 2G\boldsymbol{\alpha}^* + 2\lambda\boldsymbol{\alpha}^* - 2\boldsymbol{y}\| \\
&= \|2(\bar{G} - G)\boldsymbol{\alpha}^* + \nabla f_{rg}(\boldsymbol{\alpha}^*)\| \\
&\leq 2\sum_i |\alpha_i^*| \|\bar{G}_{\cdot,i} - G_{\cdot,i}\|_2 \\
&= 2\Delta,
\end{aligned}$$

where we use the fact that $\nabla f_{rg}(\boldsymbol{\alpha}^*) = 0$ and $\nabla \bar{f}_{rg}(\bar{\boldsymbol{\alpha}}) = 0$. Since the objective function $\bar{f}(\cdot)$ is $2\lambda$-strongly convex, we have $\|\bar{\boldsymbol{\alpha}} - \boldsymbol{\alpha}^*\| \leq \frac{\Delta}{\lambda}$, where $\lambda$ is the largest eigenvalue for $G$. □

### 7.5 Proof of Theorem 2

*Proof.* Let $f$ be the objective function of kernel SVM, $\bar{f}$ be the objective function of kernel SVM using the approximate kernel $\bar{G}$, we have

$$\begin{aligned}
\nabla \bar{f}(\boldsymbol{\alpha}^*) &= \bar{G}\boldsymbol{\alpha}^* - G\boldsymbol{\alpha}^* + G\boldsymbol{\alpha}^* - \boldsymbol{e} \\
&= (\bar{G} - G)\boldsymbol{\alpha}^* + \nabla f(\boldsymbol{\alpha}^*),
\end{aligned}$$

therefore $\|\nabla \bar{f}(\boldsymbol{\alpha}^*) - \nabla f(\boldsymbol{\alpha}^*)\| \leq \Delta$. Define $P(\boldsymbol{\alpha}) = \text{proj}_\Omega(\boldsymbol{\alpha} - \nabla f(\boldsymbol{\alpha})) - \boldsymbol{\alpha}$ be the projected gradient where $\Omega$ is the bounded constraint in the kernel SVM problem. Since $\boldsymbol{\alpha}^*$ is the optimal

solution of the kernel SVM dual problem, from the optimality condition,

$$\nabla_i f(\boldsymbol{\alpha}^*) \begin{cases} = 0 & \text{if } 0 < \alpha_i^* < C \\ \leq 0 & \text{if } \alpha_i^* = C \\ \geq 0 & \text{if } \alpha_i^* = 0 \end{cases}$$

If $\alpha_i^* = 0$, $\alpha_i^* - \nabla_i f(\boldsymbol{\alpha}^*) < 0$, so $P_i(\boldsymbol{\alpha}^*) = 0$. Similarly, we can show $P_i(\boldsymbol{\alpha}^*) = 0$ for the other two cases, and thus we have $P(\boldsymbol{\alpha}^*) = 0$.

Similarly, we have $\bar{P}(\boldsymbol{\alpha}) = \text{proj}_\Omega(\boldsymbol{\alpha} - \nabla \bar{f}(\boldsymbol{\alpha})) - \boldsymbol{\alpha}$, and we can show

$$\|\bar{P}(\boldsymbol{\alpha}^*)\| \leq \|\nabla \bar{f}(\boldsymbol{\alpha}^*)\| \tag{11}$$

by the following arguments. For each $i$, let $t_i = \alpha_i^* - \nabla_i \bar{f}(\boldsymbol{\alpha}^*)$. Consider three cases: $C \geq t_i \geq 0$, $t_i < 0$, and $t_i > C$. If $C \geq t_i \geq 0$, we have $\bar{P}_i(\boldsymbol{\alpha}^*) = t_i - \alpha_i^* = -\nabla_i \bar{f}(\boldsymbol{\alpha}^*)$. If $t_i < 0$, we have $\bar{P}_i(\boldsymbol{\alpha}^*) = -\alpha_i^*$, and thus $t_i < 0$ implies $\nabla \bar{f}(\boldsymbol{\alpha}^*) > \alpha_i^*$, so we have $|\bar{P}_i(\boldsymbol{\alpha}^*)| < |\nabla_i \bar{f}(\boldsymbol{\alpha}^*)|$. If $t_i > 0$, we can prove $|\bar{P}_i(\boldsymbol{\alpha}^*)| < |\nabla_i \bar{f}(\boldsymbol{\alpha}^*)|$ similarly.

From (11) we then have

$$\|\bar{P}(\boldsymbol{\alpha}^*)\| \leq \|\nabla \bar{f}(\boldsymbol{\alpha}^*)\| \leq \Delta.$$

We then apply the global error bound proved in [25]. They consider the function $f(\boldsymbol{x}) = g(E\boldsymbol{x}) + \boldsymbol{b}^T \boldsymbol{x}$, where $g(\cdot)$ is $\sigma$-strongly convex but $f(\boldsymbol{x})$ may not be strongly convex. They show that minimizing $f(\boldsymbol{x})$ with bounded constraint $\Omega = \{\boldsymbol{x} \mid A\boldsymbol{x} \leq \boldsymbol{d}\}$, then

$$\|\boldsymbol{x} - \bar{\boldsymbol{x}}\| \leq \theta^2 \frac{1+\rho}{\sigma} \|P(\boldsymbol{x})\|,$$

where $\bar{\boldsymbol{x}}$ is the closest optimal solution to $\boldsymbol{x}$, and $\theta$ is a positive constant independent of $\boldsymbol{x}$ (see [25] for the detailed definition).

In our case, we consider $g(C\boldsymbol{\alpha}) - \boldsymbol{e}^T \boldsymbol{\alpha}$ to be the objective function and $g(\boldsymbol{z}) = \boldsymbol{z}^T W^\dagger \boldsymbol{z}$ is positive definite, and applying the above lemma we can show (8). $\qquad \square$

### 7.6 Proof of Theorem 3

*Proof.* First, we decompose $\Delta$ by

$$\begin{aligned}
\Delta &= \sum_i |\alpha_i| \|G_{i\cdot} - \bar{G}_{i\cdot}\| \\
&= \sum_i |\alpha_i| \|G_{i\cdot} - (CW^\dagger C^T)_{i\cdot}\| \\
&\leq \sum_i |\alpha_i| \|G_{i\cdot} - Z_{i\cdot}\| + \sum_i |\alpha_i| \|Z_{i\cdot} - (CW^\dagger C^T)_{i\cdot}\|, \tag{12}
\end{aligned}$$

where $Z \in \mathbb{R}^{n \times n}$ is defined by $Z_{ij} = K(\boldsymbol{u}_{\pi(i)}, \boldsymbol{x}_j)$. We then bound the first term by

$$\begin{aligned}
\|G_{i\cdot} - Z_{i\cdot}\| &\leq \sqrt{\sum_j \left( K(\boldsymbol{x}_i, \boldsymbol{x}_j) - K(\boldsymbol{u}_{\pi(i)}, \boldsymbol{x}_j) \right)^2} \\
&\leq \sqrt{\sum_j C_K \|\boldsymbol{x}_i - \boldsymbol{u}_{\pi(i)}\|^2} \\
&= \sqrt{n C_K \|\boldsymbol{x}_i - \boldsymbol{u}_{\pi(i)}\|^2}.
\end{aligned}$$

So

$$\begin{aligned}
\sum_i |\alpha_i| \|G_{i\cdot} - Z_{i\cdot}\| &\leq \sqrt{n} \sqrt{\sum_i \alpha_i^2 \|G_{i\cdot} - Z_{i\cdot}\|^2} \\
&\leq n\sqrt{C_K} \sqrt{\sum_i \alpha_i^2 \|\boldsymbol{x}_i - \boldsymbol{u}_{\pi(i)}\|^2)} \\
&\leq n\sqrt{C_K} \sqrt{D_{\boldsymbol{\alpha}^2}(\{\boldsymbol{u}_j\}_{j=1}^m)}.
\end{aligned}$$

(a) Letter     (b) Covtype     (c) Usps

(d) Webspam     (e) Kddcup     (f) a9a

Figure 5: Comparison between our proposed method and LDKL.

Next we bound the second term of (12). We first define a matrix $Y \in \mathbb{R}^{n \times m}$ where $Y_{ij} = K(\boldsymbol{u}_{\pi(i)}, \boldsymbol{u}_j)$. Based on this definition, $Z = YW^\dagger C^T$, therefore

$$Z - CW^\dagger C^T = -(C - Y)W^\dagger C^T.$$

Then we have

$$\sum_i |\alpha_i| \|(Z - CW^\dagger C^T)_{i\cdot}\| \le \sum_i |\alpha_i| \|((C - Y)W^\dagger C^T)_{i\cdot}\|.$$

By definition,

$$(C - Y)_{ij}^2 = (K(\boldsymbol{x}_i, \boldsymbol{u}_j) - K(\boldsymbol{u}_{\pi(i)}, \boldsymbol{u}_j))^2 \le C_K \|\boldsymbol{x}_i - \boldsymbol{u}_{\pi(i)}\|^2.$$

Therefore,

$$
\begin{aligned}
\sum_i |\alpha_i| \|((C - Y)W^\dagger C^T)_{i\cdot}\| &\le \sum_i |\alpha_i| \sqrt{\sum_{j=1}^n (\|(C-Y)_{i\cdot}\| \|W^\dagger\| \|C_{j\cdot}\|)^2} \\
&\le \sum_i |\alpha_i| \sqrt{n \|W^\dagger\|^2 k \gamma_{\max} C_K \|\boldsymbol{x}_i - \boldsymbol{u}_{\pi(i)}\|^2} \\
&\le n \|W^\dagger\| \sqrt{k \gamma_{\max} C_K} \sqrt{D_{\boldsymbol{\alpha}^2}(\{\boldsymbol{u}_j\}_{j=1}^m)}
\end{aligned}
$$

$\square$

### 7.7 More Comparisons with LDKL

Figure 5 shows comparison between DC-Pred++ with LDKL on kernel SVM problem.

### 7.8 Comparison with other divide-and-conquer algorithms

We show the comparison with DC-SVM and DC-Nyström in this section. We compare the following three approaches:

- DC-SVM: the divide-and-conquer SVM proposed in [8] with the early prediction strategy. We use kmeans clustering instead of kernel kmeans in the data division step, which gives similar prediction accuracy but is much faster in terms of both training and prediction speed.
- DC-Nyström: In DC-SVM, each subproblem is solved exactly using LIBSVM. However, this leads to $O(d\bar{n})$ prediction time complexity where $\bar{n}$ is the average number of support vectors in one cluster. To speed up the prediction, a straightforward way is to replace

Figure 6: Comparison of DC-SVM, DC-Nyström, and DC-Pred++ on two real-world datasets. The results clearly show that DC-Pred++ outperforms other two algorithms in terms of prediction speed, which indicates our proposed techniques are useful.

LIBSVM by Nyström approximation approach. More precisely, We apply Nyström approximation to compute the approximate solution of each subproblem, and then solves the resulting linear SVM problem. The prediction time complexity is reduced to $O(dm)$ where $m$ is number of landmark points in each cluster.

- DC-Pred++: Our proposed algorithm – we apply pseudo-landmark points approach and weighted kmeans to select landmark points on top of DC-Nyström.

In Figure 6 we show that DC-Pred++ outperforms other algorithms, which indicates our two innovations, adding pseudo landmark points and using weighted kmeans to select landmark points, are useful for speeding up the prediction time of kernel machines.

## 7.9 Trade-off of Parameters

We study the performance of DC-Pred++ with various parameter settings on the Usps dataset. We first vary the number of clusters $k$, and choose the number of landmark points for each cluster $m_i = 0.1 s_i$ where $s_i$ is number of support vectors in that cluster. The results are presented in Table 3. When $k$ is small, both $s_i$ and $m_i$ are large, resulting in more prediction time. When $k$ is larger, the accuracy becomes worse because the algorithm ignores too much between-cluster information (the same observation was shown in [8]). The prediction time cannot be reduced to $0$ because we need $\log(k)$ inner products to determine the cluster of a testing sample. We observe our method is stable for a wide range of $k$.

Table 3: Performance of DC-Pred++ with different number of clusters $k$ on Usps dataset.

| Number of clusters ($k$) | 4 | 16 | 64 | 256 | 1024 |
|---|---|---|---|---|---|
| Prediction accuracy | 91.2% | 93.4% | 95.5% | 92.3% | 86.7% |
| Prediction time | 64x | 20x | 12x | 9x | 11x |

Next we vary number of landmark points. Fixing $k = 64$, for each cluster we choose number of landmark points $m_i = r s_i$ with different ratios $r$. The results are presented in Table 4. We can observe that the prediction time increases when the number of landmark points increases, and the prediction accuracy increases as well until $r$ is large enough.

## 7.10 Comparisons on kernel ridge regression.

Figure 7 shows comparison between DC-Pred++ with other kernel approximation based methods on kernel ridge regression problem.

Table 4: Performance of DC-Pred++ with different number of landmark points ($m_i = rs_i$), where $m_i$ is number of landmark points in the $i$-th cluster, and $s_i$ is number of support vectors in that cluster.

| $r$ | 0.05 | 0.1 | 0.2 |
|---|---|---|---|
| Prediction accuracy | 94.3% | 95.5% | 95.6% |
| Prediction time | 9x | 12x | 19x |

(a) Cpusmall

(b) Census

(c) Cadata

(d) YearPredictionMSD

(e) mnist2M

Figure 7: Kernel ridge regression results for various datasets. x-axis is the prediction cost and y-axis shows the Test RMSE. All the results are averaged over five independent runs.