[Reviews · NeurIPS 2014]

Submitted by Assigned_Reviewer_12

This paper describes some new methods for reduced kernel approximation methods (and local models) to improve the speed of kernel machines. While the paper is otherwise well written, I found the derivation of the methods to brief to be at all confident I understand what has been done (it is a bit beyond my area of expertise), so a more detailed exposition might help to make the methods more easily taken up by practitioners etc. A nice feature of the work is that the experimental evaluation suggests that the method is useful, and is well described (including the information in the supplementary information) such that it should be quite easily replicated.
Summary: This paper describes methods for improving the speed of kernel machines using reduced kernel expansions and local models. The experimental evaluation is good, but I am not confident that I have really understood what was done.

Submitted by Assigned_Reviewer_25

The paper addresses the problem of speeding up prediction of kernel methods for large-scale problems for which the models are too large and hence the prediction is too slow. The proposed approach entails construction of pseudo landmark points, in the spirit of Nystrom approximation. Unlike the true landmark points which are sampled from the data, for pseudo landmark points the values of their kernel dot products with arbitrary points can be predicted from the same values for the landmark points. Hence the increased accuracy of model approximation can be attained with the same computational complexity as using the original set of landmark points.

While the specific idea proposed in the paper is certainly novel, the overall conceptual motivation for the paper is somewhat weak in my opinion. Especially for large-scale problems, it is not the prediction but the training phase that constitutes the main computational bottleneck. It is virtually unfeasible to train large-scale models with nonlinear kernels when the kernel matrix must be stored in memory. Assuming this problem can be solved by low-rank approximation the complexity of the resulting model would then be tractable, hence I see no major gain to be made from speeding up the prediction phase. The idea of pseudo landmark points may be interesting to increase the prediction accuracy at a fixed complexity of the training model. The goal would then be not to accurately approximate the kernel matrix but rather to raise the predictive ability of the model (at fixed cost). However, the analysis provided in the paper does not go into this direction.

On the technical part, several issues are unclear:

1. No bias is considered in the SVM formulation in line 111.

2. Eq. 2: Why is G used in the definition of \bar{W}? This is counter-intuitive: we are trying to approximate G with \bar{G} depending on \bar{W} depending on G. A typo?

3. What is measured in Table 1? What is the meaning of 'x' in the values like '12.8x' etc? This table is not explained very well in the ensuing text. One can guess that the numbers pertain to some speedup, but what is the baseline for this comparison?

UPDATE:

The notation used in the paper you have referred to in the response is much more clear, and computational implications are clearly explained. Please update your notation accordingly for the sake of self-sufficiency. The 'x' notation should be clearly expalained: I have almost misinterpreted it again when reading your response.
Summary: The paper proposes a somewhat interesting idea for speeding up prediction of large-scale kernel machines but its conceptual motivation is rather weak. Several technical issues as well as experimental results are unclear.

Submitted by Assigned_Reviewer_29

This paper presents two new insight to compute efficiently a product of a kernel matrix with a vector. Empirical evidence reported on the associated implementation (called DC-pred++) demonstrates the interest of the proposed approach.

The paper is very well written, clear, well motivated and the reported empirical evidence is quite convincing, even if some reported results are not state of the art. The code used to provide results should be available on the net.

Typo
L237 weigthed → weighted
Summary: Strong paper
Author Feedback
Author rebuttal: We thank reviewers for their comments/suggestions.

In this file we use the citation format [number] to denote the references in the paper, and (names, year) to denote the new references in the end of this rebuttal file.

1. To all the reviewers:
The goal of this paper is to improve the prediction time of kernel methods, and its importance has been stated in many related papers [1,4,11,20]. The good news is that in addition to achieving state-of-the-art prediction efficiency, our algorithm also has fast training time. Table 1 in the paper shows that our method can train a model on the CoverType dataset with more than half million samples in 7 minutes, therefore we are confident that our algorithm works on large-scale datasets.

2. To Reviewer_12 and Reviewer_29: Due to space limitations, we have condensed descriptions in the submission. We expect to soon put the full paper and code online so that others can easily use the code and understand our algorithms.

3. To META reviewer: Comparisons with "post-processing techniques" and Fastfood.

1) Post processing methods:
Thanks for pointing out the line of research on "reduced-set" methods (the first two papers pointed by the META reviewer belong to this category), and we will add more discussions on that. There is a comprehensive literature review (Jung and Kim, 2014) below published this year discussing the reduced-set methods, which shows that the incremental greedy method implemented in the open source software *STPRtool* achieves the best performance. Therefore we compare with this incremental greedy approach below.

In addition, we also compare with another post-processing method ([4]). In the experiment, we fix the number of groups to be 1% of the number of support vectors as suggested by [4].

2) Fastfood: we have compared with Fastfood in the kernel ridge regression experiments (see Figure 3). For the SVM problem, LDKL has already been shown to outperform Fastfood (in LDKL's website). For clarity we also include the comparison with Fastfood below.

The tradeoff between the classification accuracy and the prediction time for STPRtool can be controlled by the number of greedy steps; for [4] it can be controlled by a parameter tau; and for Fastfood it can be controlled by number of blocks. In order to have a fair comparison, for each method we show multiple results, and we can observe that DC-Pred++ achieves the best performance. For the parameter settings in DC-Pred++, we use the same C and gamma listed in Section 7.2.

We use pairs (a%, bx) to indicate that the method achieves a% classification accuracy, and the prediction time = b * (prediction time of linear models).

Usps:

DC-Pred++:
95.56% (accuracy), 14.4x (times slower than linear SVM)

[4]:
(82%, 32.9x)
(85.6%, 75.2x)
(94.1%, 241x)

Fastfood:
(87.9%, 40x)
(90.9%, 80x)
(93.8%, 160x)

STPRtool:
(91.4%, 20x)
(93.6%, 50x)
(95.7%, 100x)

Letter:

DC-Pred++:
95.90%, 12.8x

[4]:
(90.4%, 40x)
(94.1%, 241x)
(95.9%, 2551x)

Fastfood:
(93.2%, 40x)
(94.5%, 80x)
(95.7%, 160x)

STPRtool:
(76%, 20x)
(85.9%, 50x)
(90.6%, 100x)

Webspam:

DC-Pred++:
98.4%, 20.5x

[4]:
(93%, 67x)
(96.2%, 214x)
(97.1%, 2053x)

Fastfood:
(97.0%, 40x)
(97.8%, 80x)
(98.4%, 160x)

STPRtool:
(82.4%, 20x)
(91.6%, 50x)
(95.9%, 100x)

In summary, our method is still the best and we will include the results in the final paper.

4. TO META Reviewer: Yes. For [23] we refer to the paper pointed by you.

5. To Reviewer_25: "the overall conceptual motivation for the paper is somewhat weak..."

Nystrom approximation can be used to approximate the kernel matrix and speed up kernel machines, and from Table 1 we can see that the performance is suboptimal even when rank=200 (see the 5-th column). In this case, it requires 200 inner product computations to make one prediction, which is too slow for many real-time systems (e.g., web applications, robotic applications ...). Therefore state-of-the-art Nystrom method is not good enough, and we reduce the prediction time to 10~20 inner products with a better classification accuracy, which is a big improvement.

Also, as we mentioned in the point 1 above, although we want to optimize the prediction time, our method still has fast training time. We agree that the psuedo landmark point technique can be potentially applied to speed up the training time, and it is an interesting research direction.

6. To Reviewer_25: "No bias is considered in the SVM formulation"

In the paper we ignore the bias term for simplicity, but nothing prevents us from adding it. Since our algorithm is a special case of Nystrom approximation, we can easily incorporate the bias term into the algorithm using the approach in (Zhang et al., 2012) below. In practice we found that there is no significant improvement by using the bias term, which is also observed in [9,13].

7. To Reviewer_25: "Why is G used in the definition of \bar{W}"

We want to use a low-rank approximation \bar{G} to approximate G, so when the sampled columns C from G are fixed, the optimal \bar{W} can be written in eq (2). Please see page 2732 of [27] for detail explanations.

8. To Reviewer_25: "What is the meaning of 'x'?"

The actual prediction time is normalized by the linear prediction time, so 12.8x means the actual prediction time= 12.8 * (the prediction time of linear models) = 12.8 * (time for computing one inner product). This is a typical measurement for the prediction time, which is also used in the LDKL paper.

References:

K. Zhang, L. Lan, Z. Wang, and F. Moerchen. Scaling up Kernel SVM on Limited Resources: A Low-rank Linearization Approach. AISTATS, 2012.

H. Jung and G. Kim. Support Vector Number Reduction: Survey and Experimental Evaluations. IEEE Transactions on Intelligent Transportation Systems, 2014.